# How Are Smallholder Farmers Involved in Digital Agriculture in Developing Countries: A Case Study from China

Lin Xie , Biliang Luo and Wenjing Zhong *

National School of Agricultural Institution and Development, South China Agricultural University, Guangzhou 510642, Guangdong, China; xielin@scau.edu.cn (L.X.); luobl@scau.edu.cn (B.L.)
* Correspondence: wjzhong@scau.edu.cn; Tel.: +86-138-0888-9127

**Abstract:** Digital transformation in agricultural practices may lead to a "digital divide" between small and large farms, owing to the characteristics and availability of digital technology. This paper sought to use a case study in Chongzhou County, Sichuan Province in China, to analyze how smallholder farmers in developing countries access such digital agriculture and share the benefits of digital agricultural transformation. Small farmers may own a larger scale farm through forming cooperatives; they are also indirectly involved in digital agriculture through agriculture outsourcing. The outsourcing market is expected to grow, which will allow for the evolution of a digital agricultural service platform, the development of a digital agricultural business organization consortium, and the continued expansion of a healthy digital ecology. This paper revealed important policy implications, stemming from the fact that the implementation of inclusive digital agriculture relies on two key shifts: (1) transformation from land scale operations to service scale operations and (2) from inclusive technological progress to inclusive organization innovation.

**Keywords:** smallholder farmers; digital agriculture; developing countries; China





## 1. Introduction

About three billion rural people of two-thirds of developing countries live in small farm households, working on land plots smaller than two hectares [1]. The digital transformation of agriculture has many benefits to smallholder farmers [2], including improved market transparency [3–7], enhanced farm productivity [8–11], and more efficient logistics [12,13]. In particular, under the public health crisis created by the COVID-19 pandemic, digital agriculture has helped developing countries resist the pandemic's adverse effects on food production and supply chains [14,15].

Digitalization boosts the connectivity of all actors across the agrifood system [16]. More specifically, it helps farmers access technical information, obtain higher quality seeds, collect real-time data, improve the traceability of food, and enhance their competitiveness. Therefore, it is an innovative solution to problems of the current global food system [17]. To this end, the transmission of agricultural information through mobile technologies has increased yields by 4% in sub-Saharan Africa and India, according to a meta-analysis from six studies; it has also increased the odds of adopting the recommended agrochemical inputs by 22% [18]. Given this impact, it is clear that the digital transformation of agriculture is an essential component for developing countries.

Despite its clear benefits, it is necessary to pay attention to the possible adverse effects of digital technology on smallholder farmers. With the increase of agricultural productivity, agricultural digitalization may lead to the emergence of a "digital divide" between small farms and large farms. With the commercialization of technology, it is increasingly difficult for small farmers to obtain the support of modern agricultural technology [19]. Compared with large farms, smallholders cannot afford large investments in digital agricultural technology [18]. Given this, how to better enable small farmers to access digital technology benefits is an essential—but complicated—question [20]. A recent FAO report

showed that digital agriculture in developing countries lags behind that in developed countries [21]. Real-world practice patterns have shown that the application of digital agriculture in developing countries is still at a relatively low level. For example, the main topics of concern include the role of certain communication tools (e.g., cell phones) in accessing the information on input and commodities prices [22,23]. More advanced digital technologies (e.g., big data applications) are also being discussed in the recent literature, but are occurring primarily in Europe and North America [24]. The capacity to apply digital approaches varies greatly across different countries; critically, the agricultural systems of developing countries are often not capable of digital transformation, especially in the least developed countries [23].

Recognizing that there are obstacles to the application of digital technology by small farmers, we need to answer a fundamental question: do small farmers have to be the main entities to apply such digital technology? For instance, Young [25] discussed the application of technology from the perspective of labor division, indicating that, "The mechanism of increasing returns is not to be discerned adequately by observing the effects of variations in the size of an individual firm or a particular industry, for the progressive division of labor and specialization of industries is an essential part of the process by which increasing returns are realized. What is required is that industrial operations be seen as an interrelated whole." This gives us another way to involve smallholders in digital agriculture. Through the division of labor within the agricultural industry, smallholders can be served, rather than functioning as the main entities applying digital approaches. In this way, they share in the benefits of digital agriculture.

As a developing country, China has made great efforts to promote the integration of the internet, big data, and artificial intelligence with the development of agriculture. This has been particularly apparent in rural China, where there has been vigorous development of digital agriculture. As a result, China has accumulated useful experiences with regard to integrating and implementing digital agriculture [26]. Some areas of China have established a relatively complete outsourcing service system through the division of labor within the agricultural industry. This approach has allowed them to realize the use of modern agricultural technology by small farmers [27,28].

Given this background, we sought to conduct a long-term and extensive field survey and unstructured interview in Chongzhou County, Sichuan Province in China, from 2013 to 2020. In this region, small farmers indirectly access digital agriculture through an outsourcing service system. This work analyzed the chosen case study in detail, starting from the theory of labor division. The analysis then focused on the internal mechanism behind why the social service system worked and then proposed possible solutions for the development and regulation of digital agriculture in developing countries more broadly. After this introduction, the structure of this article is as follows: (1) a literature review regarding digital agriculture and introduction of the research background for this research; (2) research area, research methods, and data collection; (3) case study description; (4) discussion, focusing on the significance of digital agriculture access for small farmers in developing countries; and (5) research conclusion.

## 2. Research Background

Digitalization, or the sociotechnical process of applying digital innovations, is an increasingly ubiquitous trend across all facets of modern life [25]. Agriculture is also transforming to digital, with the definition of digital agriculture being "the use of digital technologies, innovations, and data to transform business models and practices across the agriculture value chain" [29]. Emerging technologies such as artificial intelligence, robotics, big data, the Internet of Things, gene editing, and drones are all being used to solve challenges related to food production [30,31]. The emergence of digital technologies in food systems gives farmers many benefits. For instance, soil data have formed digital maps that help farmers apply agrochemicals in a more targeted manner [32,33]. In other applications, sensors have been used to detect plant- and plot-level soil moisture, fertilizer input, weeds,

and disease [34]. Weather forecast data have also helped farmers more accurately adjust their production decisions [33,35]. Satellite imaging has provided a wealth of crop growth data, which has improved the measurement of agricultural productivity [36]. Finally, big data approaches have been used to provide predictive insights into farming operations, driving real-time operational decisions [26].

Developed countries have invested heavily in digital agriculture. For example, the British government's GBP 4.7 billion "Industrial Strategic Challenge Fund" takes artificial intelligence and data as one of the four challenge areas and has specific plans focusing on precision agriculture [37]. In 2019, Canada pledged USD 50.3 million over five years to support agricultural strategic priority plans (CASPP), including digital agriculture. The EU also pledged to provide around EUR 100 million from 2018-2020 to fund the development and application of digital agriculture [38].

The application range of digital technology is growing rapidly. In 2015, digital technology was used to manage 65% of the arable farmland in the Netherlands, compared to only 15% in 2007 [39]. In 2015, 20% of the agricultural service providers in the United States used telematics, compared to 13% two years ago [40]. McKinsey's latest report has also pointed out that if the planned agricultural interconnections are successfully achieved, the industry will add USD 500 billion additional value to global GDP by 2030 [41].

As a developing country, China's investment in digital agriculture is lower than that in developed countries; despite this, it has created a firm foundation in digital technology and economic development. In 2018, the level of network broadband accessibility is very high, with approximately 96% of China's administrative villages having been connected to fiber-optic network by the end of 2018. Rural broadband access number increased by 23.64 million households throughout the year, reaching 117 million households [42]. Moreover, data from China Internet Network Information Center shows that the rural internet penetration rate reached 55.9% by the end of 2020. Figure 1 shows the growth of China's rural internet users and rural internet penetration rate from 2016 to 2020.

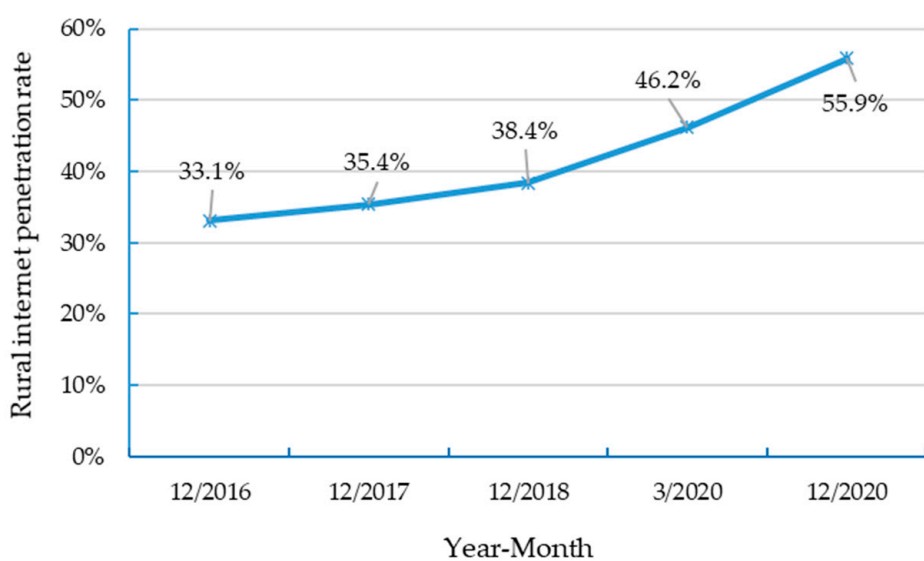

**Figure 1.** The rural internet penetration rate from 2016 to 2020.

In terms of mobile networks, the total number of mobile phone users in China reached 1.566 billion. The mobile phone penetration rate reached 112.2 units per 100 people at the end of 2018, of which 1.165 billion were 4G mobile phone users and 140 million were 3G mobile phone users. Even in rural areas, the 4G service coverage rate of administrative villages has reached 95%, meaning nearly all rural residents have access to relatively stable, high-speed mobile network services [42].

Considering the massive potential of digital technology, the Chinese government has paid great attention to the digital transformation of agriculture in recent years. In December

2019, the Ministry of Agriculture and Rural Affairs of China issued the "Digital Agriculture and Rural Development Plan (2019–2025)". This plan aims to fully promote the digital transformation of agricultural production and operation, including planting informatization, intelligent animal husbandry, fishery intelligence, seed industry digitization, the diversification of new business models, and the quality and safety control of food. A report of IFPRI shows the frequency of keywords used in national policy documents issued by the state council of China from 2006 to 2017. The results show that since 2006, the Chinese government has paid increasing attention to the development of internet access in rural areas. Since 2012, the Chinese government has also worked hard to promote the development of e-commerce in rural areas [43]. Figure 2 also shows that since 2010, the development of the agricultural internet of things has also received increasing attention.

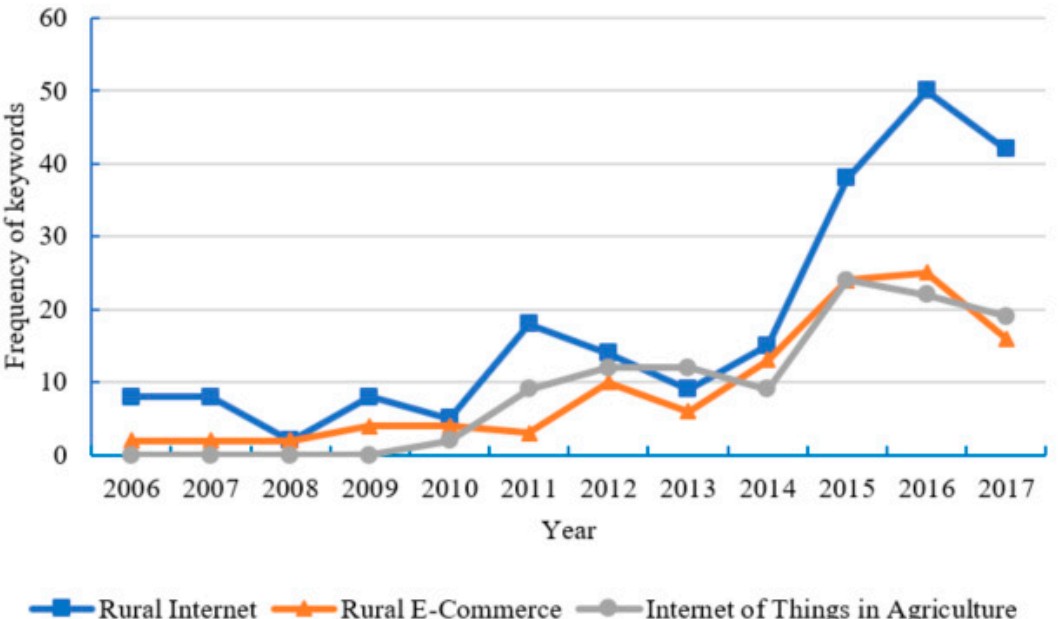

**Figure 2.** Frequency of key words used in national policy documents issued by the state council of the PRC., 2006–2017.

At present, China's digital agriculture focuses on two aspects: first, rural e-commerce on the consumer side, using a network platform to expand the sales market of Chinese agricultural products. According to recent statistics, in 2018, China's rural e-commerce market reached RMB 175 billion, increasing to RMB 187.36 billion in the first half of 2019. This represents a year-on-year increase of 25.3% [44]. Second, China's digital agriculture has focused on smart agricultural production that applies modern digital technologies (e.g., the Internet of Things, remote sensing). This approach uses sensors and software to control agricultural production through mobile terminals and/or computer platforms. According to recent estimates, China's smart agriculture's potential market size is expected to grow from USD 13.7 billion in 2015 to USD 26.8 billion in 2020 [45]. However, the digitization level of China's agriculture remains lower than in other industries and fields. In 2018, China's agricultural digital economy only accounted for 7.3% of the agricultural added value, which was far below the 18.3% and 35.9% of the manufacturing and service industries, respectively. This low level of agricultural digitization is partially explained by problems like technology mismatch with China's agricultural reality [44]. All of these constitute the background of this case study.

## 3. Study Area and Research Methodology

### 3.1. Study Area

Sichuan Province is one of the largest agricultural provinces in China and rice has been planted in the region for over two thousand years. Given this history, it has always

been one of the most important crop-producing areas in China. The study area in this research is Chongzhou County, a suburban county of Chengdu, which is the capital city of Sichuan Province. The study area was located at the junction of the eastern plains and western mountains in Sichuan Province. The land area of Chongzhou county is 1089 square kilometers, of which the plain area accounts for 52% of the total area, the mountainous area accounts for 43%, and the hilly area accounts for 5%. In 2019, the total agricultural output value of Chongzhou County was RMB 3.94 billion, accounting for 10.338% of the total regional GDP, and the annual grain planting area reached 467,000 mu(mu is equivalent to 0.0667 ha) with a total grain output of 218,000 tons; the oil planting area was 173,000 mu with a total output of 27,000 tons [46].

Chongzhou County is a typical rural area in China based in its characteristics of topography, industry, and population structure, and its development is influenced by rapid industrialization and urbanization. Chongzhou County is a major agricultural county and an important exporter of rural labor. In 2019, the rural population of Chongzhou County was 446,603, with only 86,772 individuals working in agriculture [46]. A large number of rural laborers have flowed into the industry and service sectors. Owing partly to this outflow of labor, Chongzhou County has achieved a very high level of agricultural mechanization. In 2019, the total power of agricultural machinery in Chongzhou County reached 420,800 kilowatts, with 70 large- and medium-sized tractors, 6363 small tractors, and 967 combine harvesters. Moreover, in 2019, the annual mechanical farming area was 42,802 hectares, the electromechanical irrigation area was 45,287 hectares, the mechanical sowing area was 34,602 hectares, the mechanical harvest area was 38,402 hectares, and the agricultural mechanization rate reached 96% of Chongzhou County [46]. Within the context of smallholder farming in China, Chongzhou County's successful adoption of modern agricultural production technology has attracted the attention of researchers.

With the continuous development of the agricultural outsourcing services, some service providers in Chongzhou County have begun to build platforms that gather various services. Ultimately, this will introduce digital technology transaction platforms to guide Chongzhou County's agriculture towards digital transformation. Chengdu, where Chongzhou County is located, has a good foundation for an agricultural and rural informatization infrastructure. For instance, Chengdu's village's optical fiber coverage and wireless communication network coverage have both reached 100%, and its mobile communication 4G network coverage ranks second in China. Moreover, Chengdu is one of China's first 5G cities. There are other important indicators for this foundation, including its leading enterprises in agricultural industrialization, the existence of specialized farmers' cooperatives and family farms, and the presence of other new agricultural business entities (e.g., computer application rate); moreover, the network coverage rate and smartphone coverage rate of employees have all reached 99% [47]. For these reasons, we chose Chongzhou County as the study area for this research. Figure 3 shows the location of Chongzhou County in China.

*3.2. Research Methodology*

This study used qualitative research methods, focusing on in-depth field interviews, literature, and archive research. From 2013 to 2020, we conducted a long-term and extensive field follow-up survey of organizations and individuals involved in applying digital agricultural technology in Chongzhou County, including government officials at the county and town levels, digital agricultural platform companies, cooperatives, agricultural service organizations, and ordinary farmers. We conducted the field surveys five times, each lasting an average of four days. The field surveys mainly included unstructured interviews, group discussions, participant observations, material and information selection, and cooperative case analysis. The interviewees and interview content in group discussions and unstructured interviews are shown in Table 1. We recorded or made notes for all interviews. In group interviews, we usually conducted in two phases: first, discussions with government officials for about two hours; second, discussions with the chief

operation officer (COO) of the farmland cooperatives, agricultural service organizations, and small farmers for about three hours. To reduce the possibility of interruption during the interviews, in the second phase, it was suggested that local government officials were to be avoided. The objects of the unstructured interviews were the key persons mentioned in the interview. The unstructured interview usually lasted for one to two hours and might be extended to online interviews or phone interviews in follow-up surveys when necessary. Our follow-up interviews comprised more than 20 county-level or town-level officials, two digital agriculture platform company managers, five cooperatives' managers, three service organization leaders, and approximately 30 smallholder farmers, with a total number of more than 60 interviewees. After the surveys, we obtained three main types of information: records of the interviews, the archives of the cooperatives (e.g., cooperatives' financial accounts and book sheets from 2015 to 2020), and the government statistical yearbooks, including cooperative bulletins and other statistical sources.

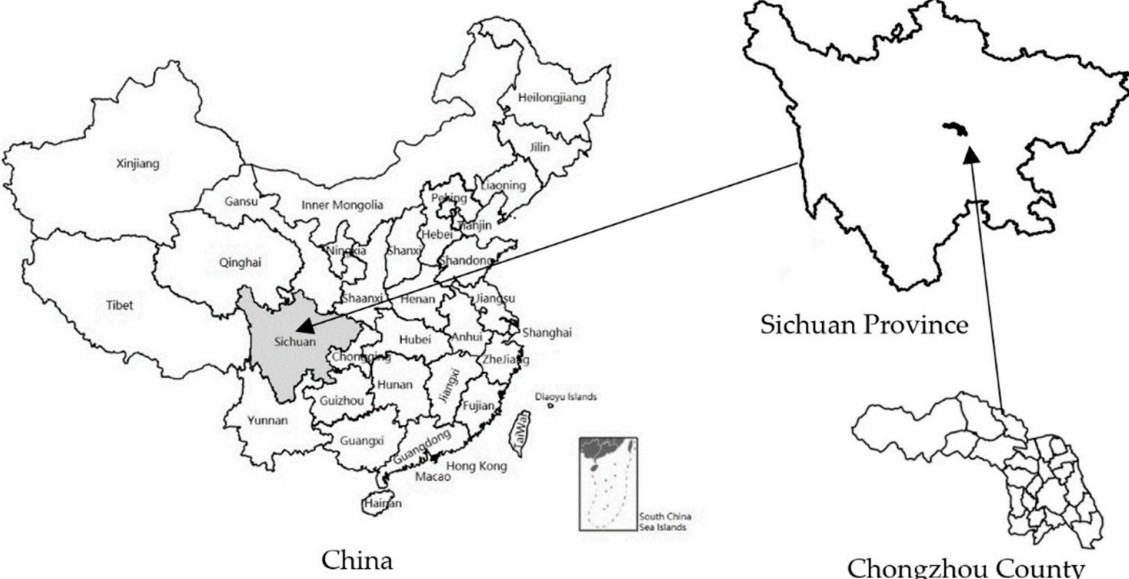

**Figure 3.** Location of the study area.

We conducted interviews to identify the process of smallholders' digital transformation and its influencing factors to verify the authenticity of the interview content through the identification of archive materials. The use of multiple respondents mitigates the potential biases of any individual respondent by allowing the information to be confirmed by several sources [48,49]. The use of multiple respondents also enables inducing richer and more elaborate models because different individuals typically focus on complementary aspects of major decisions [50,51]. At this stage, the questions concentrated on facts, events, and direct interpretations, rather than hearsay or vague commentary [52]. Overall, we can obtain a richer, mutually verified, and relatively accurate understanding of the phenomenon after integrating interview data and archive data. Before the analysis, we made sure that there were no theoretical preferences and prior hypotheses. We used tables and graphs to facilitate the studies [53]. The goal was to identify the theoretical constructs, relationships, and longitudinal patterns with respect to the research questions [54] and then complete the theoretical model's development.

**Table 1.** Interviewees and interview content.

| Type of Interviewees | Number of Interviewees | Interview Content |
|---|---|---|
| Government officials in the agricultural department | More than 20 | ◆ Agricultural policy implementation performance and planning design priorities;<br>◆ Problems, difficulties, and countermeasures in the transformation of modern agriculture;<br>◆ The role of farmland cooperatives and agricultural service organizations in the development of modern agriculture;<br>◆ Knowledge and understanding of digital agriculture; estimation of the development of digital agriculture. |
| Farmland cooperative COO | 5 | ◆ Farmland cooperative's establishment date, method, rules, and regulations, operation performance;<br>◆ Interest relationships with farmers, agricultural service organizations, governments, and digital platform companies;<br>◆ Business with digital platforms, knowledge and understanding of digital agriculture, and judgment of its application and development. |
| Agricultural service organization COO | 3 | ◆ Agricultural services types, scale, and performance;<br>◆ The use of digital technology in the service process;<br>◆ Business dealings with digital platforms, knowledge and understanding of digital agriculture, and the judgment of its application and development. |
| Small-scale farmer | More than 30 | ◆ Characteristics of individual and family;<br>  - If farming, topics are about the farming scale, outsourcing services, usage of digital agriculture technology, and attitudes or opinions of digital agriculture.<br>  - If a member of a farmland cooperative, topics are about the benefit-sharing mechanism, degree of satisfaction, plans, etc. |
| Manager and technician of the digital agriculture platform company | 2 | ◆ The business model of the digital platform;<br>◆ The service types, performance, and relationship with relevant stakeholders of the digital platform;<br>◆ Difficulties encountered in the promotion of digital platforms and judgements on future development. |

## 4. Case Description

*4.1. Agricultural Organization: Formation of Cooperatives and Centralized Land Management*

Historically, Chongzhou County is an agricultural area dominated by small farmers; given this, it is challenging to apply advanced agricultural technology to the county's agricultural production and the production efficiency is low. In 2010, an attempt was

made to solve this problem when 30 of the 33 households in the 15 groups of Liba Village, Longxing Town registered and established the first farmland shareholding cooperative in Chongzhou County with a price of RMB 900 per mu. This cooperative was named the Yangliu Land Contract Management Cooperative (hereafter referred to as "Yangliu Co-operative"). The Yangliu Cooperative followed the principles of voluntary membership, freedom of withdrawal, and benefit/risk-sharing. The farmers used the land they owned to convert their individual shares into cooperative shares. The farmland shareholding cooperative then elected both a Board of Directors and a Board of Supervisors to undertake the decision-making and supervision functions. At the beginning of its establishment, the Yangliu Cooperative's shareholding farmland was 101.27 mu. Although it is far from the size of the large farms found in the United States, its scale is already very impressive in Sichuan, which is still dominated by small farmers.

Cooperative members without large-scale farming experience can hardly match the abilities of a farm with a scale of 100 mu or more. Given this, the Chongzhou County Rural Development Bureau hired a technician as the manager of the cooperative. According to the government officials' description, the technician was a young man with a major in agronomy, he also had local rural life experience, and practical agriculture skills. With the government's working experience, he possessed more knowledge about the agricultural policies and market information to carry out large-scale operations. Similar to the CEO of an enterprise, the manager is responsible for the operating activities of the cooperative. The manager brings scarce human capital for operation and management to the cooperative and improves the cooperative's economic benefits, such as increasing grain income and developing cash crop cultivation.

We interviewed the first manager of the Yangliu Cooperative. He believed that there were many benefits for farmers to became cooperative members. First, the cooperatives had more substantial investment capabilities and higher levels of management. The income of the members included land rental income and dividends, which was much higher than self-farming income. Second, as the manager was responsible for agricultural production, the members could seek off-farm opportunities and earn wage income. Third, members could withdraw freely, so it had a binding effect on the cooperative and the manager. By joining the cooperative, most farmers stated that the land rental income and dividends were higher than their original farming income, and the labor force of the family could earn additional off-farm income via working in the city. If they didn't want to work in the city, farmers could set up a small-scale aquaculture or be hired by the cooperatives to earn wages. Compared to working in the city, starting a small business at home or working for a cooperative could keep the farmers' rural lifestyles and fulfill their emotional needs.

After years of development, smallholder farmers in Chongzhou County formed 246 farmland shareholding cooperatives, operating across a land area of 316,000 mu and an average operating area of 1284.6 mu. By the end of 2018, the proportion of farmers participating in the cooperative exceeded 80% [28].

### 4.2. Deepening the Division of Labor in Agriculture: Agricultural Outsourcing Services and the Application of Agricultural Machinery

Studies have shown that owing to their inherently small scale of operations, small-holder operations hinder the application of modern agricultural technology [55,56]. The establishment of the Chongzhou County Cooperative has expanded its management scale to a more considerable extent. However, the scale is still not large enough to encourage the consumption of agricultural machines. This is because the high productivity of agricultural machinery means that the farming tasks can be completed more quickly, so the machinery becomes an idle asset.

Young [25] pointed out that the application of new technologies should be considered from an industry perspective, not just from within the organization. After developing the cooperative in Chongzhou County, managers found that compared to before, the expansion of the farms' scale was hard to match with artificial farming. It was also not possible to complete all of the production tasks by managers. This situation drove demand for

agricultural outsourcing services. Taking the Yangliu Cooperative as an example, at the beginning of its establishment, the managers searched for agricultural outsourcing service providers to provide machinery services. This would have enabled the Yangliu cooperative to achieve farming mechanization.

Another example is the local Yunfeng Farm, which purchased many batches of agricultural machinery. At the end of 2012, the farm had machinery and equipment valued at RMB 2 million, which provides mechanical services for 4000–5000 mu of farmland. The COO of Yunfeng Farm told us that they mainly served large-scale farms. They only served small-scale farmers if time permitted and the price was higher. As the farmland usage was still for agricultural purposes, the government didn't interfere with cooperatives' product diversity. However, agricultural services centralized production types; thus, service prices decreased, and service levels increased. This process is the so-called service specialization leading to the concentration of production types. The concentration of production contributes to a larger scale of service market and promotes further development of service market [57]. In 2011, Yunfeng Farm's turnover exceeded RMB 2 million. The large market scale was so attractive that many of their relatives and friends came to the farm and tried to learn how to use the agricultural machinery.

Cooperatives achieve agricultural mechanization by hiring outsourcing services, which then stimulates market investment in agrarian service. Ultimately, this deepens the division of labor and self-reproduction. With the development of agricultural outsourcing service providers, the outsourcing services market gradually expand to the businesses of agricultural technology consulting, agricultural mechanization, agricultural material distribution, professional seedling, pest control, field transportation, and grain drying and storage. These services are like "nanny" services for agricultural production entities and include the entire process of agricultural production. At the end of 2019 in Chongzhou County, the amount of agricultural land benefiting from agricultural machinery services reached 300,000 mu, and the coverage of farmers reached 98% [46]. In the past few years, Yunfeng Farm continued to develop and become a cooperative. By 2020, the Yunfeng Cooperative had 85 agricultural pieces of machinery (sets) of various types, providing agricultural services for more than 30,000 mu, covering seven counties (cities, districts), including Chongzhou.

### 4.3. Digital Transformation of Agriculture: Integration of Digital Agricultural Service Platforms

As the demand for outsourcing services increased in Chongzhou County, a mismatch between demand and supply appeared in its agricultural outsourcing service transaction market. The seasonal characteristics of agriculture were prominent, so an agricultural service provider would often receive far more orders than its capacity in a short period. For those agricultural service providers who had received orders that exceeded their service capacity, orders were referred to other agricultural service providers with surplus production capacity. With the increasing frequency of agricultural service exchange activities, some service providers gradually transferred to specialists to exchange agricultural services and established a service supermarket specializing in providing agricultural service intermediary services.

In March 2012, Chengdu Shunonghao Agriculture Co., Ltd. was established. It is a service supermarket that had built a platform to integrate various service providers, marked service prices clearly, and provided a one-stop service. The emergence of the service supermarket had dramatically reduced the search cost of the business entities. Requiring only a phone call, the service supermarket would coordinate for customers and provided door-to-door service of agricultural services as soon as possible. At the end of August 2013, Chengdu Shunonghao Agriculture Co., Ltd. built six agricultural service supermarkets in Qiquan Town, Longxing Town, and Jixie Town in Chongzhou County. First, it integrated 22 large or medium-sized agricultural machinery service providers, which collectively had 320 sets of large- and medium-sized agricultural machinery and 662 professional employees. Second, it integrated six labor service providers with more than

1000 employees. Third, it integrated 16 plant protection professional service providers and more than 700 sets of plant protection machinery. Moreover, it integrated two professional seedling raising service providers, built a 2000-ton grain drying warehouse, and explored "grain bank" services [58].

The most important factor restricting the deepening of the division of labor is coordination cost [59]. In Chongzhou County, the establishment of the service supermarket was a revolution, which significantly reduced the coordination cost of agricultural services, greatly expanded the scale of the market, and promoted the further deepening of the division of labor in agriculture. With the application of digital technology, Chongzhou County began to use digital technology to transform and upgrade service supermarkets. In 2019, Sinochem Agriculture built a Modern Agriculture Platform (MAP) in Chongzhou County together with the BeiDou navigation system, China Meteorology, and other systems. The platform contained a precision planting system, digital operation system, and quality control traceability system. The system had been called the Space–Air–Ground Integrated Digital Agricultural Service Platform. The system relied on remote sensor monitoring, precision weather, AI recognition, online technology live broadcasts, and other technologies. Ultimately, this approach had allowed for the timely diagnosis of pests and diseases, as well as accurate services for agricultural operations. The platform had also been used in the digital transformation of services, such as seed selection, fertilizer distribution, plant protection, an inspection of grain purchasing, and storage and market information. Figure 4 shows the four phases of the digital transformation of smallholder farming in Chongzhou county.

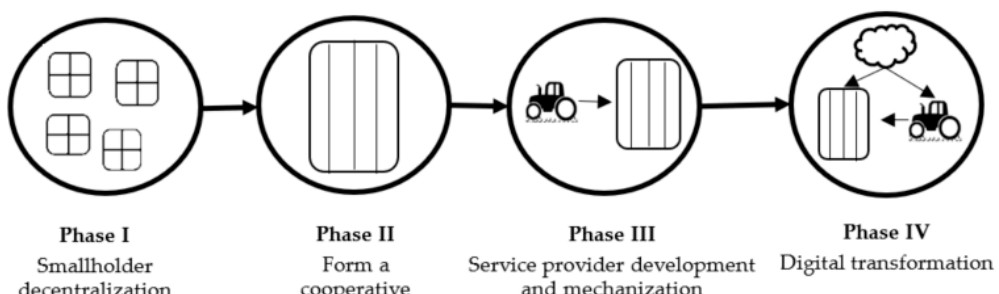

**Figure 4.** Four phases of the digital transformation of smallholder farming in Chongzhou county.

Chongzhou County's digital agricultural service platform's operation mode is shown in Figure 5, where the dotted line refers to the information flow. The data platform obtains market data from the finance system, inputs and products' markets, remote sensing data from satellites, and operating data from service providers' smart agricultural machinery and smart drones and soil sensing and weather station data on the cooperative farmland. These data are then integrated and analyzed using big data, resulting in assessments regarding task allocation services for the service providers and technical guidance services for the cooperatives. After receiving the platform's orders, the service providers offer different services to the cooperative according to the platform's instructions. Although the digital agricultural service platform is commercial, the government does not interfere with the daily operation of the platform. However, for supervision purposes, the government has the right to access the data of the agricultural service platform, so the data are also accumulated in the government's memory. By the end of 2019, Chongzhou County's digital agricultural service platform provided services to service providers and cooperatives across 18 counties and in 7 cities or prefectures in Sichuan Province, covering a total of 170,000 mu of farmland.

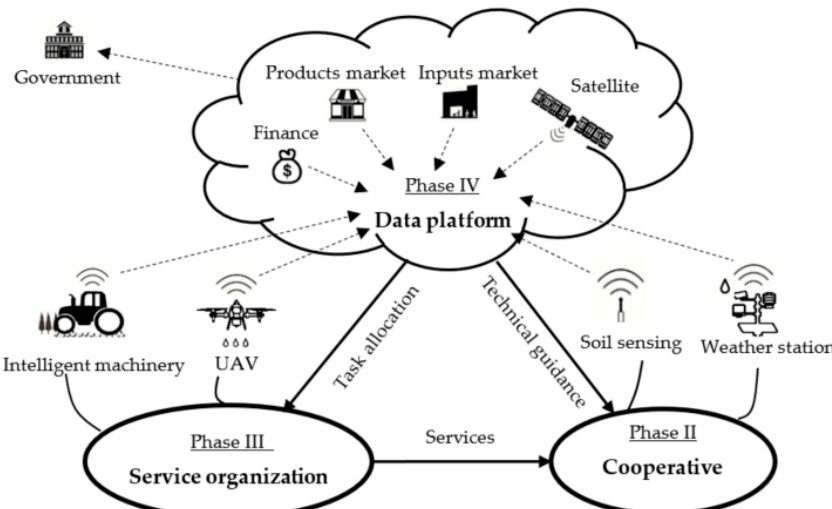

**Figure 5.** Chongzhou county's digital agricultural service platform. Note: The dotted line refers to the information flow.

The digital agricultural service platform integrates funds, technology, brand information, and other elements to improve resource allocation efficiency. It integrates cooperatives and agricultural service providers into a whole data system and establishes a close alliance of the agricultural operation organizations centered on the platform. The platform helps to match transactions between the cooperatives and the service providers, and reduces the transaction costs of the agricultural services. The platform also significantly improves cooperatives' efficiency for field inspections, which ultimately saves money. As mentioned above, by 2019 the Yangliu Cooperative owned farmland totaling 3850 mu. The managers supervise seedlings, diseases, and insects through the app developed by the digital agricultural service platform. In particular, during the COVID-19 epidemic, cooperative managers have relied on the app to monitor the growth of farm crops, providing a guarantee for the regular progress of the agricultural activities. In this regard, officials, platform managers and technicians all believed that with the accumulation of data, MAP would play a more important role and lead the future development of modern agriculture. Figure 6 shows the app of digital agriculture service platform (MAP).

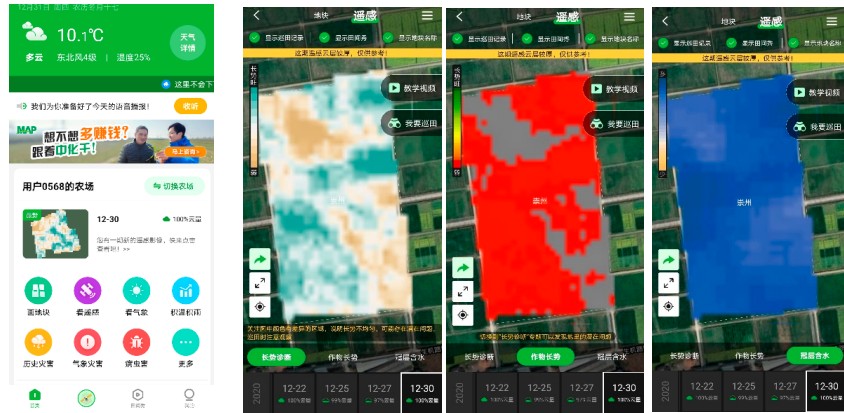

**Figure 6.** App of digital agriculture service platform (MAP). Note: The first picture is the entry screen of the app, and the next three pictures show how to monitor the growth of crops.

## 5. Discussion

Many developed countries have large-scale agricultural farms. For example, in the United States in 2019, the average farm area reached 444 acres [60]. "Large-scale farms" refers to a farm with plenty of land, and it is a concept compared to smallholder farms,

which are short of land or have around two hectares of land, as defined by FAO. In fact, according to the data from The National Bureau of Statistics of China, the average farm size in China is 0.43 ha, and the size of 92.49% of them is less than 2 ha (30 Mu). However, the service for the land scale required by digital technology is usually larger than 2 ha. For example, the digital agricultural service platform in Chongzhou County requires a scale of more than 50 mu (3.3 ha). So these large-scale farms have natural advantages in applying digital technology, and it is complex and difficult to develop digital technology that is inclusive to small farmers [19]. As a result, there is a problem of an agricultural "digital divide". Of the approximately 570 million farms in the world, more than 475 million are smaller than two hectares [61]. In the context of the digital transformation of agriculture, the "digital divide" will widen the already existing gap in agricultural productivity between developing and developed countries. This is especially realistic since developing countries are usually dominated by smallholder farmers [21].

It is likely that the direction of how digital technology's development is frustrating to small farmers. In some developing countries, agricultural policies often encourage small farmers to transfer farmland to create large-scale farms, thereby increasing agricultural productivity. For example, in China, there is a view to strengthen Chinese agriculture's competitiveness, and it is necessary to promote large-scale farms through land transfer [62]. However, compared with the United States, China has a large agricultural population and an imperfect social security system, which determines that China's agricultural production cannot rely on large-scale farms. In fact, in 2018, China's total agricultural land transfer accounted for 57.17% of the inflow of rural households, while only 10.31% were transferred to the enterprises with scale management ability [63]. In the short term, the practice of transferring and expanding farm scale is challenging to achieve.

However, in the case of Chongzhou County in Sichuan province of China, we have shown a possible path for smallholder farmers in developing countries to be involved in the development and implementation of digital agriculture through an organized way. First, establishing cooperatives allows for larger land operations. Second, agricultural mechanization is achieved by supporting the development of agricultural service providers. Third, a digital agricultural business organization consortium is formed by building a digital agricultural service platform, which evolves into a healthy, digital ecology. Collectively, this allows for improved agricultural productivity.

Given what was done in Chongzhou County, future policy should emphasize two shifts: (1) scale the economy of the land shifts to scale economy of the agricultural service; (2) inclusive technological innovation shifts to the inclusiveness of the organizational innovation. According to the website of the Ministry of Agriculture and Rural Affairs of China, the number of agricultural outsourcing service organizations in China will exceed 900,000 by the end of 2020; moreover, the agricultural production service area will exceed 1.6 billion mu, of which the food crop area will exceed 900 million mu. Agricultural production service will involve 70 million rural households [64]. Taken together, past agricultural development has already shown that the "two shifts" will have a national significance and are not just unique to Chongzhou County.

However, the digital transformation of agriculture will still bring a series of problems and challenges. First, digital agriculture reduces physical labor and improves agricultural life. However, manual labor, traditional farming lifestyles, and rich agricultural production experiences are of great significance for farmers to participate in and lead to a better understanding of their land and environment [65]. The use of digital technology may lead to the marginalization of the experience of agricultural production and a disconnection between farmers and agriculture [66], which brings challenges to the inheritance of the traditional agricultural culture [67]. Second, in the context of agricultural digitization, machinery makes autonomous decisions based on big data without manual intervention; in other words, machines replace labor. However, in China, agriculture still provides jobs for 194.45 million people [68], and remains an available job for individuals who have failed to enter urban occupations [69]. Therefore, we should pay attention to the social and

political implications of digital agriculture transformation. Third, digitization often means using large amounts of data. Consequently, privacy and security must be fully protected. In China, most farmers are not aware of this problem. However, the online equipment required for digital agriculture may open exposure to cyber threats. For example, if there are loopholes in the network security firewall settings, third parties may access sensitive data, steal data, and/or destroy equipment. Finally, when compared to small farmers and other entities with weak abilities to defend their rights, large companies with more negotiating power are more be able to integrate their data and make decisions based on the information [70–72].

## 6. Conclusions

As digital agriculture technology is more friendly to large-scale farms, the digital transformation of small-scale farms in developing countries has encountered significant challenges. This research discussed how small farmers might be involved in digitalization from mechanization. By analyzing the case study presented by Chongzhou County, Sichuan province in China, small farmers obtained a relatively large farmland scale by forming cooperatives. Then they outsourced agricultural production to service providers, who provided mechanized agricultural services to cooperatives. Some service providers have developed trading platforms and digitized them to create an integrated digital agricultural service platform. This platform integrates farm management systems, precision planting systems, digital operation systems, and quality control traceability systems. Ultimately, this would allow for the digital transformation of the entire industrial chain of agriculture production.

The critical findings of this research for the digital transformation of small farmers in developing countries are as follows: first, focus on the cultivation and development of outsourcing service systems is critical, rather than focusing on the scale of farmland. Second, when it is challenging to achieve inclusive digital technology progress, we should focus on inclusive organizational innovation to digitalize small farmers. Of course, this study also pointed out that the digital transformation of Chongzhou County's agriculture still had many problems, including changing the agricultural production ecology, replacing labor, agricultural data privacy and security, and the political and social ramifications of such a transformation.

The shortcomings of this research are as follows: as a case study, this research has not quantitatively evaluated the impact of digital agriculture on agricultural production or smallholders' income; this research outlined a conceptual diagram of the process and operation mode of smallholders' involvement in digital agriculture, but it has not collected data for quantitative analysis of the position and role of each participant in the digital ecology. Solving the above two problems is meaningful for future research.

**Author Contributions:** Conceptualization, L.X.; methodology, L.X., W.Z., and B.L.; formal analysis, L.X. and W.Z.; data curation, L.X.; writing—original draft preparation, L.X. and W.Z.; writing—review and editing, L.X., W.Z., and B.L.; supervision, L.X. All authors have read and agreed to the published version of the manuscript.

**Funding:** This research was funded by the Philosophy and Social Science Foundation of Guangdong Province, grant number GD19CYJ15"; National natural science foundation of China, grant number 71703041; Philosophy and Social Science Planning Project of Guangzhou, grant number 2020GZYB33, 2020GZGJ73.

**Institutional Review Board Statement:** Not applicable.

**Informed Consent Statement:** Informed consent was obtained from all subjects involved in the study.

**Data Availability Statement:** The data presented in this study are available on request from the corresponding author.

**Acknowledgments:** The authors are grateful to the reviewers and the editor for their extensive and constructive review comments, which helped improve the quality of this paper.

**Conflicts of Interest:** The authors declare no conflict of interest.

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
