# Peer review of "How Are Smallholder Farmers Involved in Digital Agriculture in Developing Countries: A Case Study from China"

_land, doi:10.3390/land10030245_

Round 1
Reviewer 1 Report
The aim and topic of the article are interesting, but the manuscript needs improvement.
Main comments to the Authors:
- The description of the research methodology needs to be improved. In particular, the scope of the field surveys should be described. It is also unknown what the purpose of the long-term and extensive field follow-up survey (unstructured interviews) of organizations and individuals involved.
- In the Case description and Conclusions sections, reference should be made to field surveys (interviews) and their results should be presented.
Author Response
Comment 1: The description of the research methodology needs to be improved. In particular, the scope of the field surveys should be described.
Authors’ Response: Thank you for pointing this out. We have carefully discussed the scope of the field surveys in research methodology section (Line 236-245 in the highlight revision):
The interviewees and interview content in group discussions and unstructured interviews are shown in Table 1. We recorded or made notes for all interviews. In group interviews, we usually conducted in two phases: first, discuss with government officials for about 2 hours; second, discuss with the chief operation officer (COO) of farmland cooperatives, agricultural service organizations, and small farmers for about 3 hours. To reduce the possibility of interruption during the interviews, in the second phase, local government officials were suggested to be avoided. The objects of unstructured interviews were key persons mentioned in the interview. The unstructured interview usually lasted for 1 to 2 hours and might be extended to online interviews or phone interviews in follow-up surveys when necessary.
Comment 2: It is also unknown what the purpose of the long-term and extensive field follow-up survey (unstructured interviews) of organizations and individuals.
Authors’ Response: Thank you for pointing this out. We add Line 259-272 in the highlight revision to discuss the purpose of the long-term and extensive field follow-up survey,” the purpose is to identify the process of smallholders’ digital transformation and its influencing factors and to verify the authenticity of the interview content through the identification of archive materials”. In Line 260-273, we add a description of the case study method and indicate the basis for our approach:
Use of multiple informants mitigates the potential biases of any individual respondent by allowing information to be confirmed by several sources [48-49]. The use of multiple informants also enables inducing richer and more elaborated models because different individuals typically focus on complementary aspects of major decisions [50-51]. At this stage, the questions concentrated on facts, events, and direct interpretations, rather than hearsay or vague commentary [52]. Overall, we can get a richer, mutually verified, and relatively accurate understanding of the phenomenon after integrating interview data and archive data. Before the analysis, we made sure that there were no theoretical preferences and prior hypotheses. We used tables and graphs to facilitate studies [53]. The goal was to identify the theoretical constructs, relationships, and longitudinal patterns with respect to the research questions [54] and then complete the theoretical model's development.
Comment 3: In the Case description and Conclusions sections, reference should be made to field surveys (interviews) and their results should be presented.
Authors’ Response: In the section of Case description, we have added content from interview material as follows:
(1) Interviews with the government officials' (Line 293-297 in the highlight revision):
According to the government officials' description, the technician was a young man with a major in agronomy, and he had local rural life experience, and had practical agriculture skills. With the government's working experience, he owned more knowledge about the agricultural policies and market information to carry out large-scale operations.
(2) The first manager of Yangliu Cooperative (Line 302-314 in the highlight revision):
We interviewed the first manager of Yangliu Cooperative. He believed that there were many benefits for farmers to become cooperative members. First, cooperatives had more substantial investment capabilities and higher levels of management. The income of members included land rental income and dividends, which was much higher than self-farming income. Second, as the manager was responsible for agricultural production, the members could seek off-farm opportunities and earn wage income. Third, members could withdraw freely, so it had a binding effect on cooperative and manager. By joining the cooperative, most farmers stated that the land rental income and dividends were higher than their original farming income, and the labor force of the family could earn additional off-farm income via working in the city. If they didn’t want to work in the city, farmers could set up a small-scale aquaculture or be hired for cooperatives to earn wages. Compared to working in the city, starting a small business at home or working for a cooperative could keep the farmers' rural lifestyle and fulfill their emotional needs.
(3) The COO of Yunfeng Farm (Line 346-356 in the highlight revision):
The COO of Yunfeng Farm told us that they mainly served large-scale farms. They only served small-scale farmers if time permits and the price is higher. As the farmland usage is still for agricultural purposes, the government didn’t interfere with cooperatives' product diversity. However, agricultural services centralize production types; thus, service prices decrease, and service levels increase. This process is the so-called service specialization leading to the concentration of production types. The concentration of production contributes to a larger scale of service market and promotes further development of service market [57]. In 2011, Yunfeng Farm's turnover exceeded RMB 2 million. The large market scale was so attractive that many of their relatives and friends came to the farm and tried to learn how to use agricultural machineries.
(4) The platform manager (Line 453-456 in the highlight revision):
In this regard, officials, platform manager and technicians all believed that with the accumulation of data, MAP would play a more important role and lead the future development of modern agriculture.
Reviewer 2 Report
Dear Authors,
I had the opportunity to review the manuscript ID land-1107521 title „How Smallholder Farmers are Involved in Digital Agriculture in Developing Countries: A Case Study from China”.
I agree with the topicality of the research topic: it is very important to explore the factors that can help involve smallholder farmers in digital agriculture.
Below are comments for the manuscript.
Line 383-384: What do you mean by 'large-scale' farm? From what size? The average farm size doesn’t always refer to the land use structure. For example, in Hungary, based on the data of the Farm structure survey 2016, the average size of the agricultural land was 12.8 hectares, which, according to the logic of the authors, suggests a small-scale (smallholder) farming system. The actual land use in Hungary is quite concentrated: 66.6% of the farms own less than 2 hectares, and the agricultural area they cultivate is 2%. The number of farms with an area of at least 50 hectares is 4.4% of the number of farms in Hungary, and the agricultural area they use is 74.4%.
Line 264: word repetition (County)
In Table 1: Average number of members (number)
The results evaluated by the authors and the conclusions drawn from them are, in my opinion, correct.
Author Response
Comment 1: Line 383-384: What do you mean by 'large-scale' farm? From what size? The average farm size doesn’t always refer to the land use structure. For example, in Hungary, based on the data of the Farm structure survey 2016, the average size of the agricultural land was 12.8 hectares, which, according to the logic of the authors, suggests a small-scale (smallholder) farming system. The actual land use in Hungary is quite concentrated: 66.6% of the farms own less than 2 hectares, and the agricultural area they cultivate is 2%. The number of farms with an area of at least 50 hectares is 4.4% of the number of farms in Hungary, and the agricultural area they use is 74.4%.
Authors’ Response: Thank you for your Comment. In this paper, large-scale farms refer to a farm with plenty of land, and it is a concept compared to Smallholder farm, which is short of land or has land around 2 hectares as defined by FAO. In fact, according to the data from The National Bureau of Statistics of China, the average farm size in China is 0.43 ha, and 92.49% of farms is less than 2 ha (30 Mu). However, the service for the land scale required by digital technology is usually larger than 2 ha. For example, the digital agricultural service platform in Chongzhou County requires a scale of more than 50 mu (3.3 ha). We have marked the above content in the form of footnotes (in the Discussion section).
Comment 2: Line 264: word repetition (County)
In Table 1: Average number of members (number)
The results evaluated by the authors and the conclusions drawn from them are, in my opinion, correct.
Authors’ Response: Thank you for pointing those out. Indeed, it is our oversight. We have corrected those errors.
Reviewer 3 Report
The background of this publication is the fact that digitalisation is increasingly finding its way into agriculture. As this is associated with considerable costs, its application is beneficial to large farms. Small farms, on the other hand, can profit little from its advantages. Therefore, the application of digital technologies between large and small farms is becoming more and more divergent. Using a case study in Sichuan province, the authors aim to show how smallholders can be involved in the use of digital technologies.
Since smallholder farmers are reported to make up most agricultural producers, and they also form large parts of the population in many developing countries, the topic is extremely relevant. Simultaneously, however, the interrelationships between agricultural production, the development of adapted technologies, the changing dynamic market conditions and the social consequences are incredibly complex. In this respect, the case study methodology used and the qualitative procedures are, in my opinion, appropriate to this complexity. For the transfer of the results obtained to other regions, however, the challenge also arises that the concrete contexts must be described in detail. The recent experiences should at least be hinted.
The authors use an expansive definition of the term digital agriculture. This means that very different information supply functions, decision support, automation and marketing can be addressed. This can result in various benefits for small farmers. For me, the paper describes comparatively relatively little about the concrete benefits that small farmers can derive from the offers. It would be helpful if these service functions were described in more detail. The fact that China is already more advanced in introducing such technologies in the small-scale agricultural sector and has a good infrastructure makes the case study valuable for developments in other regions.
Data and information for the study were obtained from secondary data (for example, cooperatives' records). The qualitative interviews conducted interviews with smallholder farmers, service providers, cooperative managers, and public administration. This provides the opportunity for a dense description of the case study situation. The paper gives the impression that the possibilities have not been fully exploited, however.
The results highlighted by the authors, in particular, show that before developing adapted digital solutions to problems, the prerequisites for an extensive application must first be created. From this, the authors conclude that small farms must first be consolidated intensive cooperation. This opens up the possibility for service providers to make functions available the small enterprises at higher costs can only provide that. The development of efficient service companies then also creates the opportunity to develop and expand digital service provision. These must then be adapted to the needs of small farmers.
Overall, the work is worth publishing. However, to better assess the relevance and transferability of the results of the case study, it would be good to add some information and make minor corrections:
- It has not become entirely clear whether the merging of the cultivated areas allows for a wide diversity of production or making standardisation necessary.
- It would help know the service companies' legal status (private sector, state funding?) and who operates the digital agricultural service platform and perhaps also finances its development (fully private sector, private-public funding). What does the wording mean that the agricultural production service covers 70 million rural households? It sounds like it is from start-ups of small farms.
- If so many interviews have been done with actors in the system, it would also help determine if and how the situation has changed from the point of view of small farmers and whether it has led to an improvement in competitiveness.
- Is it correct to write that the regions are exporters of labour?
Author Response
Comment 1: The background of this publication is the fact that digitalisation is increasingly finding its way into agriculture. As this is associated with considerable costs, its application is beneficial to large farms. Small farms, on the other hand, can profit little from its advantages. Therefore, the application of digital technologies between large and small farms is becoming more and more divergent. Using a case study in Sichuan province, the authors aim to show how smallholders can be involved in the use of digital technologies.
Since smallholder farmers are reported to make up most agricultural producers, and they also form large parts of the population in many developing countries, the topic is extremely relevant. Simultaneously, however, the interrelationships between agricultural production, the development of adapted technologies, the changing dynamic market conditions and the social consequences are incredibly complex. In this respect, the case study methodology used and the qualitative procedures are, in my opinion, appropriate to this complexity. For the transfer of the results obtained to other regions, however, the challenge also arises that the concrete contexts must be described in detail. The recent experiences should at least be hinted.
Authors’ Response: Chongzhou County is a typical rural area in China based on its topography, industry, and population structure characteristics. So, we have added a basic description of the above information in the section of Study area”, Chongzhou County is a typical rural area in China based in its characteristics of topography, industry, and population structure, and its development is influenced by rapid industrialization and urbanization”(Line 194-196 in the highlight revision). Besides, we also discussed it in the discussion section. As the Chinese government attaches great importance to agricultural services in rural areas, the agricultural services market developed rapidly nationwide. So, it is practical to copy the mode of Chongzhou County to other areas (Line 495-502 in the highlight revision).
Comment 2: The authors use an expansive definition of the term digital agriculture. This means that very different information supply functions, decision support, automation and marketing can be addressed. This can result in various benefits for small farmers. For me, the paper describes comparatively relatively little about the concrete benefits that small farmers can derive from the offers. It would be helpful if these service functions were described in more detail. The fact that China is already more advanced in introducing such technologies in the small-scale agricultural sector and has a good infrastructure makes the case study valuable for developments in other regions.
Authors’ Response: Thank you for your Comment. In the article, we believe that digital agriculture is difficult to serve small farmers directly. But through engaged in land shareholding cooperatives, small farmers would benefit from digital transformation. In Line 303-314 in the highlight revision, we have added content about the benefits of cooperatives for small farmers:
First, cooperatives had more substantial investment capabilities and higher levels of management. The income of members included land rental income and dividends, which was much higher than self-farming income. Second, as the manager was responsible for agricultural production, the members could seek off-farm opportunities and earn wage income. Third, members could withdraw freely, so it had a binding effect on cooperative and manager. By joining the cooperative, most farmers stated that the land rental income and dividends were higher than their original farming income, and the labor force of the family could earn additional off-farm income via working in the city. If they didn’t want to work in the city, farmers could set up a small-scale aquaculture or be hired for cooperatives to earn wages. Compared to working in the city, starting a small business at home or working for a cooperative could keep the farmers' rural lifestyle and fulfill their emotional needs.
Comment 3: Data and information for the study were obtained from secondary data (for example, cooperatives' records). The qualitative interviews conducted interviews with smallholder farmers, service providers, cooperative managers, and public administration. This provides the opportunity for a dense description of the case study situation. The paper gives the impression that the possibilities have not been fully exploited, however.
Authors’ Response: In the section of Case description, we have added content from interview material as follows:
(1) Interviews with the government officials' (Line 293-297 in the highlight revision):
According to the government officials' description, the technician was a young man with a major in agronomy, and he had local rural life experience, and had practical agriculture skills. With the government's working experience, he owned more knowledge about the agricultural policies and market information to carry out large-scale operations.
(2) The first manager of Yangliu Cooperative (Line 302-314 in the highlight revision):
We interviewed the first manager of Yangliu Cooperative. He believed that there were many benefits for farmers to become cooperative members. First, cooperatives had more substantial investment capabilities and higher levels of management. The income of members included land rental income and dividends, which was much higher than self-farming income. Second, as the manager was responsible for agricultural production, the members could seek off-farm opportunities and earn wage income. Third, members could withdraw freely, so it had a binding effect on cooperative and manager. By joining the cooperative, most farmers stated that the land rental income and dividends were higher than their original farming income, and the labor force of the family could earn additional off-farm income via working in the city. If they didn’t want to work in the city, farmers could set up a small-scale aquaculture or be hired for cooperatives to earn wages. Compared to working in the city, starting a small business at home or working for a cooperative could keep the farmers' rural lifestyle and fulfill their emotional needs.
(3) The COO of Yunfeng Farm (Line 346-356 in the highlight revision):
The COO of Yunfeng Farm told us that they mainly served large-scale farms. They only served small-scale farmers if time permits and the price is higher. As the farmland usage is still for agricultural purposes, the government didn’t interfere with cooperatives' product diversity. However, agricultural services centralize production types; thus, service prices decrease, and service levels increase. This process is the so-called service specialization leading to the concentration of production types. The concentration of production contributes to a larger scale of service market and promotes further development of service market [57]. In 2011, Yunfeng Farm's turnover exceeded RMB 2 million. The large market scale was so attractive that many of their relatives and friends came to the farm and tried to learn how to use agricultural machineries.
(4) The platform manager (Line 453-456 in the highlight revision):
In this regard, officials, platform manager and technicians all believed that with the accumulation of data, MAP would play a more important role and lead the future development of modern agriculture.
Comment 4: The results highlighted by the authors, in particular, show that before developing adapted digital solutions to problems, the prerequisites for an extensive application must first be created. From this, the authors conclude that small farms must first be consolidated intensive cooperation. This opens up the possibility for service providers to make functions available the small enterprises at higher costs can only provide that. The development of efficient service companies then also creates the opportunity to develop and expand digital service provision. These must then be adapted to the needs of small farmers.
Overall, the work is worth publishing. However, to better assess the relevance and transferability of the results of the case study, it would be good to add some information and make minor corrections:
It has not become entirely clear whether the merging of the cultivated areas allows for a wide diversity of production or making standardisation necessary.
Authors’ Response: Thank you for pointing this out. We have added content as follows,” As the farmland usage is still for agricultural purposes, the government didn’t interfere with cooperatives' product diversity. However, agricultural services centralize production types; thus, service prices decrease, and service levels increase. This process is the so-called service specialization leading to the concentration of production types. The concentration of production contributes to a larger scale of service market and promotes further development of service market (Luo, 2017)” (Line 348-354 in the highlight revision).
Comment 5: It would help know the service companies' legal status (private sector, state funding?) and who operates the digital agricultural service platform and perhaps also finances its development (fully private sector, private-public funding).
Authors’ Response: The agricultural service platform in Chongzhou is running by Sinochem, which merged Syngenta, as a for-profit project. Because both the central and local government is concerned about the development of digital agriculture, Chongzhou funded the company with public funds to set up its service platform in Chongzhou. Although the establishment of the service platform is mainly for commercial purposes, it is inevitable to be interfered by the government, and the government also has the right to require accessing platform data. Therefore, we added the role of government to the information flow in Figure 5.
Comment 6: What does the wording mean that the agricultural production service covers 70 million rural households? It sounds like it is from start-ups of small farms.
Authors’ Response: The Chinese government has recognized the significant role of agricultural services in applying modern agricultural technology and supports the development of agricultural service organizations through public investment. These agricultural service organizations are both from private sectors and public sectors. According to data from the Ministry of Agriculture and Rural Affairs of China, at the end of 2020, China has 900,000 service organizations that can provide agricultural services to 70 million farmers. It means that not only in Chongzhou but nationwide, it is possible to implement digital agricultural transformation through service organizations as intermediaries.
Comment 7: If so many interviews have been done with actors in the system, it would also help determine if and how the situation has changed from the point of view of small farmers and whether it has led to an improvement in competitiveness.
Authors’ Response: Our observation shows that digital agriculture is beneficial to individual small farmers. For example, small farmers can use mobile phones or computers to inquire about agricultural information, and government-funded projects can also push production information to small farmers. However, small farmers tend to be excluded from other more complex digital technologies due to capacity constraints. But through organized cooperatives, smallholder farmers can be involved in these complex digital technologies to increase agricultural productivity and reduce the level of chemical inputs. These gains will eventually become farmers’ annual dividends.
Comment 8: Is it correct to write that the regions are exporters of labour?
Authors’ Response: In China, with the development of urbanization and industrialization, many rural labors move to urban areas for employment opportunities, so rural areas are exporters of labor. At the same time, the eastern and southern regions of China have more developed industries, while our study area is in the relatively undeveloped western region, many local rural labors have temporarily migrated to eastern and southern China. Therefore, we consider the description of "the study region is an exporter of labor " is credible.
Reviewer 4 Report
- The title reflects the contents of the paper.
- The abstract of the paper reads okay
- Introduction section motivates the reader to the subject of the paper
- In paragraph 4.3, in the information flow, the authors are not referred to the eGovernment services. Are there any digital agricultural eGovernement services? Please check:
Gerald Glenn F Panganiban (2019) E-governance in agriculture: digital tools enabling Filipino farmers, Journal of Asian Public Policy, 12:1, 51-70, DOI: 10.1080/17516234.2018.1499479
Maria Ntaliani, Constantina Costopoulou, Sotirios Karetsos, Efthimios Tambouris, Konstantinos Tarabanis, Agricultural e-government services: An implementation framework and case study, Computers and Electronics in Agriculture, Volume 70, Issue 2, 2010, Pages 337-347,
Bournaris, T. Evaluation of e-Government Web Portals: The Case of Agricultural e-Government Services in Greece. Agronomy 2020, 10, 932. https://doi.org/10.3390/agronomy10070932
5. Conclusions section should be developed to highlight the unique contributions of the paper, limitations of the research and some future research directions.
Author Response
Comment:
- The title reflects the contents of the paper.
- The abstract of the paper reads okay
- Introduction section motivates the reader to the subject of the paper
- In paragraph 4.3, in the information flow, the authors are not referred to the eGovernment services. Are there any digital agricultural eGovernement services? Please check:
Gerald Glenn F Panganiban (2019) E-governance in agriculture: digital tools enabling Filipino farmers, Journal of Asian Public Policy, 12:1, 51-70, DOI: 10.1080/17516234.2018.1499479
Maria Ntaliani, Constantina Costopoulou, Sotirios Karetsos, Efthimios Tambouris, Konstantinos Tarabanis, Agricultural e-government services: An implementation framework and case study, Computers and Electronics in Agriculture, Volume 70, Issue 2, 2010, Pages 337-347,
Bournaris, T. Evaluation of e-Government Web Portals: The Case of Agricultural e-Government Services in Greece. Agronomy 2020, 10, 932. https://doi.org/10.3390/agronomy10070932
- Conclusions section should be developed to highlight the unique contributions of the paper, limitations of the research and some future research directions.
Authors’ Response: Thank you for your Comment. To promote the application of digital agricultural, the local government has funded the platform. Although the digital agricultural service platform is commercial, the government does not involve the platform's daily operation. Still, for supervision, the government has the right to obtain the agricultural service platform's data and store the data in the government's memory. So, we added the role of government in Figure 5 and explained its function.
Besides, we added limitations of the research and some future research directions at the end of the article.
Round 2
Reviewer 1 Report
The revised version of the manuscript is definitely better. The article can be published.